# Orphan Nuclear Receptor Nur77 Mediates the Lethal Endoplasmic Reticulum Stress and Therapeutic Efficacy of Cryptomeridiol in Hepatocellular Carcinoma

**DOI:** 10.3390/cells11233870

**Published:** 2022-12-01

**Authors:** Xudan Li, Quancheng Chen, Jie Liu, Shenjin Lai, Minda Zhang, Tidong Zhen, Hongyu Hu, Xiang Gao, Alice S. T. Wong, Jin-Zhang Zeng

**Affiliations:** 1State Key Laboratory of Cellular Stress Biology, Fujian Provincial Key Laboratory of Innovative Drug Target Research, School of Pharmaceutical Sciences, Faculty of Medicine and Life Sciences, Xiamen University, Xiamen 361102, China; 2Xingzhi College, Zhejiang Normal University, Lanxi 321004, China; 3School of Biological Sciences, University of Hong Kong, Pokfulam Road, Hong Kong, China

**Keywords:** sesquiterpenoid, drug target, orphan nuclear receptor Nur77, ER stress, mitochondrial membrane potential, hepatocellular carcinoma

## Abstract

Hepatocellular carcinoma (HCC) commonly possesses chronical elevation of IRE1α-ASK1 signaling. Orphan nuclear receptor Nur77, a promising therapeutic target in various cancer types, is frequently silenced in HCC. In this study, we show that cryptomeridiol (Bkh126), a naturally occurring sesquiterpenoid derivative isolated from traditional Chinese medicine *Magnolia officinalis*, has therapeutic efficacy in HCC by aggravating the pre-activated UPR and activating the silenced Nur77. Mechanistically, Nur77 is induced to sense IRE1α-ASK1-JNK signaling and translocate to the mitochondria, which leads to the loss of mitochondrial membrane potential (Δψm). The Bkh126-induced aggravation of ER stress and mitochondrial dysfunction result in increased cytotoxic product of reactive oxygen species (ROS). The in vivo anti-HCC activity of Bkh126 is superior to that of sorafenib, currently used to treat advanced HCC. Our study shows that Bkh126 induces Nur77 to connect ER stress to mitochondria-mediated cell killing. The identification of Nur77 as a molecular target of Bhk126 provides a basis for improving the leads for the further development of anti-HCC drugs.

## 1. Introduction

Orphan receptor Nur77 (also known as NR4A1, TR3, or NGFI-B) belongs to the nuclear receptor superfamily (NRs) [1,2]. It is widely distributed but is expressed in considerably low abundance in normal tissues. Interestingly, it can quickly respond to various stress stimuli and is suggested to be a potential stress sensor [3,4]. Nur77 displays both tumor-promoting and -suppressing effects, which are determined by its subcellular localization [5,6]. Nucleus-localized Nur77 manifests oncogenic potential due to its inducing tumor-promoting genes, such as Cyclin D1 and VEGF [7,8]. The overexpression of Nur77 also limits CART T-cell function [9]. In contrast, targeting Nur77 nuclear export can be lethal and tumor suppressive. Such a unique anti-cancer model has been largely exploited for drug screening and discovery [2,10,11]. The beneficial Nur77 exporters identified include many natural products, such as cardenolide ATE-i2-b4 from the stem of *A. toxicaria* Lesch and hellebritoxin H-9 from the skin of toad [12]. When translocating to the mitochondria, Nur77 is demonstrated to modulate mitochondrial biology or initiate mitochondrial-dependent cell death [6,13]. Nur77 can be also directed to the endoplasmic reticulum (ER), playing a role in stress resolution or mediating autophagic cell death [3,4]. However, how Nur77 orchestrates the signaling cross-talk between the organelles to mediate its anti-cancer activity remains poorly understood.

The ER is critical for responding to intrinsic and extrinsic stressors. ER stress leads to the activation of the unfolded protein response (UPR) through three ER transmembrane proteins, PERK, IRE1α, and ATF6 [14,15]. These pathways collaboratively keep cellular homeostasis and maintain cells in a healthy status. However, such a beneficial mechanism is often hijacked by cancer cells to support their aggressive growth [16]. Most of malignant tumors are evolved to endure high levels of ER stress. Thus, the UPR is frequently upregulated and hyperactivated in a wide range of malignancies, including HCC [17]. The altered expression of PERK, IRE1α, ASK1, and other downstream molecules ss suggested to be involved in the process of nonalcoholic steatohepatitis (NASH), cirrhosis, precancerous lesions, and eventually HCC, suggesting that the UPR is extensively elevated during hepatocarcinogenesis [18,19,20]. Pre-activated UPR signaling in HCC may provide an intervenable target for combating HCC.

Interfering with ER homeostasis has been demonstrated to be an exciting field for developing novel anti-cancer drugs. Here, we identified cryptomeridiol (Bkh126; (1R,4aR,7R,8aR)-7-(2-hydroxypropan-2-yl)-1,4a-dimethyl-2,3,4,5,6,7,8,8a-octahydronaphthalen-1-ol) as a new ER perturbing agent. Cryptomeridiol belongs to the sesquiterpenoid family, which includes rich therapeutic leads for drug discovery [21,22]. It is a natural product isolated from *Magnolia officinalis* Rehd. et Wils. (*M. officinalis*), which is an ancient traditional Chinese herbal medicine widely used in folk medicine for more than 2000 years and indexed in Chinese pharmacopoeias (2020 Edition). Cryptomeridiol and several other herbal extracts of *M. officinalis* possess potent pharmacological activities against various human diseases, including anti-tumor, anti-anxiety, anti-bacterial, and liver protection activity, which await to be explored as modern drugs [23,24]. In this study, we found that cryptomeridiol targets Nur77 to sense and connect ER stress to mitochondrial dysfunction for cell killing, demonstrating an interestingly novel mechanism of action against HCC.

## 2. Materials and Methods

### 2.1. Reagents and Antibodies

Goat anti-rabbit and anti-mouse secondary antibodies conjugated to horseradish peroxidase were obtained from Thermo Fisher Scientific (Dreieich, Germany); anti-Nur77 (#3960S) and IRE1α (#3294) were obtained from Cell Signaling Technology (Danvers, MA, USA); anti–poly (ADP ribose) polymerase (PARP) (66520-1-Ig), anti-GAPDH (60004-1-Ig), anti-tubulin (66031-1-Ig), anti-p-ASK1 (Ser966) (28846-1-AP), and anti-ASK1 (67072-1-Ig) were obtained from Proteintech (Rosemont, Inc, USA); anti-p-p38 (Tyr 182) (sc-166182), anti-p38 (sc-81621), anti-p-JNK (Thr 183 and Tyr 185) (sc-6254), anti-JNK (sc-7345), anti-Bcl-2(sc-509), anti-Myc (9E10; sc-40), and anti-GFP (sc-9996) were obtained from Santa Cruz Biotechnology (Santa Cruz, CA, USA); anti-Ser/Thr (ab17464) and anti-IRE1α (phospho S724) (ab124945) were obtained from Abcam. ER stress inhibitor 4-PBA (HY-A0281), IRE1α inhibitor KIRA (HY-19708), ASK1 inhibitor GS-444217 (HY-100844), JNK inhibitor SP600125 (HY-12041), ATF6 inhibitor AEBSF (HY-12821), PERK inhibitor GSK2606414(HY-18072), and p38 inhibitor SB203580 (HY-10256) were purchased from MedChemExpress (Monmouth Junction, NJ, United States (MCE)). PEI was obtained from Invitrogen; a cocktail of proteinase and phosphatase inhibitors was obtained from Roche; enhanced chemiluminescence (ECL), protein A/G agarose Scientific, MTT, 4, 6-Diamidino-2-phenylindole (DAPI), Annexin V-EGFP/PI Apoptosis Detection Kit, anti-cy3 (SA00009-1) from Protein-tech; TUNEL Apoptosis Detection Kit (Alexa Fluor 594) were obtained from Yeasen Biotechnology (Shanghai) Co., Ltd. (Shanghai, China).

### 2.2. Isolation and Identification of Cryptomeridiol (Bkh126)

The cortex of *Magnolia officinalis* Rehd. et Wils. var. biloba Rehd. et Wils. was collected from Sanming, Fujian Province, China. In total, 10 kg of dried bark was exhausted with methanol reflux for 3 times, 4 h each time. After filtration, the extracting solution was combined and concentrated to dryness, yielding ~900 g of methanol extract. The methanol extract was suspended in distilled water and was then extracted successively with hexane, ethyl acetate, and n-butanol. The ethyl acetate extract was fractionated using silica gel column chromatography to 25 fractions by gradient elution with a solution of chloroform and methanol in ratios from 100:1 to 1:1. Cryptomeridiol (namely, Bkh126) was isolated from the 17th fraction using silica gel column chromatography eluted with chloroform and methanol (20:1–5:1 gradient elution). The structure was identified with 1H-NMR and ^13^C-NMR spectrum data analyses. Purified Bkh126 was (1R,4aR,7R,8aR)-7-(2-hydroxypropan-2-yl)-1,4a-dimethyl-2,3,4,5,6,7,8,8a-octahydronaph-thalen-1-ol (Figure 1A) [25], with a purity of ~95%.

### 2.3. Cell Culture and Transfection

Hep3B (20170315-37) and Huh-7 (20170224-05) were obtained from Cell Research and authenticated using an STR analysis (https://www.zqxzbio.com/, accessed on 15 February 2017). HepG2 (ATCC HB-8065) and HEK293T (ATCC CRL-11268) were originally obtained from American Type Culture Collection (ATCC; Manassas, VA, USA). The newly received cells were expanded, and aliquots of less than ten passages were frozen in liquid nitrogen. Cells were cultured at 37 °C in a humidified atmosphere of 5% CO_2_ in DMEM (Gibco, Grand Island, NY, USA) supplemented with 10% FBS and 1% penicillin and streptomycin (Gibco, Grand Island, NY, USA). PEI was used to transfect the cells (Invitrogen, Carlsbad, CA, USA).

### 2.4. Crispr/Cas9-Mediated Gene Knockout

We used the online CHOPCHOP website (http://chopchop.cbu.uib.no/) to design specific sgRNA sequences targeting Nur77 and Bax. The primers for Lenti-sgNur77 were as follows: forward, CAC CGT CCG AAC AGA CAG CCT GAA G; reverse, AAA CCT TCA GGC TGT CTG TTC GGA C. The primers for Lenti-sgBax were as follows: forward, CAC CGG TTT CAT CCA GGA TCG AGC A; reverse, AAA CTG CTC GAT CCT GGA TGA AAC C. The primers for Lenti-sgControl were as follows: forward, ACG GAG GCT AAG CGT CGC AA; reverse, TGC CTC CGA TTC GCA GCG TT. The plasmids were constructed and packaged into HEK293T cells (ATCC CRL-11268) using lentivirus vector lenti-CRISPR V2. The lentivirus was then used to infect HCC cells. After 48 h, puromycin was used for cell selection. Single-cell-derived colonies were expanded for 14 more days.

### 2.5. Cell Apoptosis Assays

The cells were harvested and washed twice with precooled PBS and stained with Annexin V-FITC and propidium iodide (PI) at room temperature in the dark for 15 min. Flow cytometry was used to count apoptotic cells (Thermo, Attune NxT, Dreieich, Germany).

### 2.6. Western Blotting

Cells were harvested and lysed in lysis buffer (150 mM NaCl, 1% NP-40, and 50 mM Tris–HCl (pH 7.5)) containing protease inhibitors (Sigma, Singapore). The cell lysates were separated using 10% SDS-PAGE and transferred to PVDF membranes. The membranes were blocked with 5% skim milk in TBST (10 mM Tris-HCl (pH 8.0), 150 mM NaCl, and 0.1% Tween 20). Primary antibodies were used at 1:1000 dilution and secondary antibodies conjugated with horseradish peroxidase at 1:5000 dilution. The immunoreactive bands were detected with ECL using a biomolecular imager after the final wash (Biorad, CA, USA). The blot intensities were calculated using GAPDH as a loading control.

### 2.7. Immunohistochemistry

Cells mounted on glass slides were fixed for 5 min with 4% paraformaldehyde in PBS, pH 7.4, and permeabilized for 15 min with 0.1% Triton X-100 and 0.1 mol/L glycines. The slides were blocked for 30 min at room temperature with 1% bovine serum albumin in PBS; then, they were stained for 1 h at room temperature with anti-Nur77 antibody at 1:300 dilution, followed by incubation with secondary antibodies conjugated with FITC and Cy3 (1:200 dilution). To visualize the nuclei, the cells were co-stained with DAPI (0.1 g/mL). The images were captured using a fluorescent microscope (Carl Zeiss, Oberkochen, Germany) or a confocal laser scanning microscope system (LSM-510) (Carl Zeiss, Oberkochen, Germany).

### 2.8. Co-Immunoprecipitation Assays

Transfected cells were lysed in 500 μL of IP lysis buffer (10 mM Tris–HCl (pH 8.0), 150 mM NaCl, 10 mM ethylenediaminetetraacetic acid, and 0.2% NP-40) containing protease inhibitors for the co-immunoprecipitation assay (Sigma, Singapore). The lysates were incubated with 1 μg of anti-GFP monoclonal antibody (Santa Cruz Biotechnology, Dallas, TX, USA) at 4 °C for 2 h. Immuno-complexes were then precipitated with 40 μL of protein A/G-sepharose (Santa Cruz Biotechnology, Dallas, TX, USA). Beads were washed extensively with IP lysis buffer and analyzed using Western blotting.

### 2.9. Measurement of Intracellular ROS Levels

The intracellular ROS levels were determined by assaying the extent to which 2′,7′-dichlorofluorescein-diacetate (DCFH-DA) oxidized to fluorescent dichlorofluorescein (DCF) using Reactive Oxygen Species Assay Kit (Beyotime Biotechnology, Shanghai, China). Briefly, the cells were seeded in 6-well plates and exposed to various concentrations of diethyltryptamine for varying time intervals. The cells were incubated with DCFH-DA for 30 min at 37 °C and analyzed with flow cytometry at 488 nm excitation and 525 nm emission (Thermo, Attune NxT).

### 2.10. Measurement of Mitochondrial Membrane Potential (MMP) with Fluorescent JC-1

Cells were treated with Bkh126 in combination with various kinase inhibitors. The cells were collected and resuspended in DMEM, followed by incubation with JC-1 at room temperature for 20 min. The cells were examined with a fluorescent microscope or a flow cytometer (Thermo, Attune NxT) (Carl Zeiss, Oberkochen, Germany).

### 2.11. Animal Experiments

Male BALB/c nude mice were injected subcutaneously with HepG2 cells (2 × 10^6^ cells) in the posterior flanks. The treatment of mice was initiated on day 3 post-transplantation; mice were injected intraperitoneally with vehicle or Bkh126 (20 or 40 mg/kg) once daily. Tumor volume was measured with a caliper every three days. The mice were sacrificed after a 3-week treatment. Tumors and organ tissues were collected.

### 2.12. Ethics Statement

The animal study protocols were approved by Institutional Animal Care and Use Committee of Xiamen University.

### 2.13. Statistical Analysis

Student’s tests in GraphPad Pro software (GraphPad, San Diego, CA, USA) were used to analyze the statistical significance among sets of data. Results are expressed as means ± s.d. of more than 3 experiments. Differences were considered to be significant at *p* < 0.05.

## 3. Results

### 3.1. Bkh126 Induces HCC Cell Apoptosis Dependent on Nur77 Induction

We established a natural product library for screening and identifying potential leads for drug discovery. Since Nur77 represents a promising potential drug target, we looked into those hits capable of modulating Nur77 expression and function. As a result, we found that cryptomeridiol (Bkh126; (1R,4aR,7R,8aR)-7-(2-hydroxypropan-2-yl)-1,4a-dimethyl-2,3,4,5,6, 7,8,8a-octahydronaphthalen-1-ol; Figure 1A), a kind of sesquiterpenoid obtained from *Magnolia officinalis*, and some other sesquiterpenoid compounds, such as T7, obtained from marine *P. indica*, could strongly induce Nur77 expression in various HCC cells, including HepG2, Huh-7, and Hep3B (Figure 1B). The induction of Nur77 occurred as early as 1 h, and it returned to basal level 6 h after treatment (Figure 1C). The knocking out of Nur77 strikingly impaired the apoptosis-promoting effect of Bkh126, demonstrating that Nur77 is critical for the anti-HCC activity of Bkh126 (Figure 1D).

### 3.2. Bkh126-Induced Apoptosis Is Associated with Its Activation of IRE1α and ASK1

In our pilot experiments, we showed that several sesquiterpenoid derivatives could initiate tumor cell apoptosis through inducing extensive ER stress. In addition, the chronic activation of IRE1α and ASK1 has been implicated in HCC [26,27,28,29]. We, thus, focused on dissecting whether the anti-HCC activity of Bkh126 was associated with the aggravation of the pre-elevated IRE1α-ASK1 signaling. Indeed, Bkh126 could dose-dependently stimulate IRE1α phosphorylation without affecting its total protein expression in HepG2, Huh-7, and Hep3B cells (Figure 2A). Bkh126 treatment significantly upregulated Bip and CHOP expression (Appendix A). We then determined whether other ER stress pathways were also possibly involved. The result showed that Bkh126 could promote PERK phosphorylation but did not significantly affect ATF6 expression (Appendix A). Thus, various blockers against ER stress and individual pathways were used in the following experiments. We showed that Bkh126-induced apoptosis could be abrogated by about 80–90% in the presence of ER stress inhibitor 4-PBA in HepG2, Hep3B, and Huh-7 (Figure 2B; Appendix A). IRE1α inhibitor KIR6A impaired the apoptotic effect of Bkh126 by 73–75% (Figure 2B). The inhibition of PERK with GSK2606414 also rescued the cells by about 25–30%, while the use of ATF6 inhibitor AEBSF did not induce significant change (Appendix A). These results suggest that the anti-HCC activity of Bkh126 is initiated largely depending on the activation of IRE1α. Consistently, IRE1α inhibition by KIR6A resulted in the suppression of PARP cleavage by Bkh126 (Figure 2C).

Since ASK1 mediates IRE1α signaling, we explored the role of ASK1 and its downstream signaling in the apoptotic action of Bkh126. Our results showed that Bkh126 could dose-dependently promote the activity of ASK1, followed by increased phosphorylation of JNK and p38, two substrates and effectors of ASK1 (Figure 3A). When the activity of IRE1α was blocked by KIRA6, the levels of p-ASK1, p-JNK, and p-p38 were all inhibited (Figure 3B). Biologically, Bkh126-induced PARP cleavage was reversed when the activity of ASK1 was inhibited by GS-444217 (Figure 3C).

### 3.3. Bkh126 Induces Nur77-Mediated ROS Production and Mitochondrial Dysfunction

Bkh126-induced ER stress was accompanied by increased ROS production in a dose-dependent manner (Figure 4A), which could be impaired by 4-PBA (ER stress inhibitor) (Figure 4B) or by the knocking out of Nur77 (Figure 4C), suggesting that ER disturbance and Nur77 induction mediated the effect of Bkh126 on ROS production. Bkh126-induced ROS production was associated with mitochondrial dysfunction as evidenced by the fact that Bkh126 treatment significantly altered the mitochondrial membrane potential (Δψm), which we assessed using flow cytometry and a fluorescent microscope with JC-1 staining (Figure 4D,E). JC-1 is localized within the mitochondrial matrix and forms J-aggregates to emit red fluorescence. When J-aggregates are decomposed into monomers upon Δψm loss, the emitted fluorescence shifts from red to green [30]. JC-1 staining showed that Bkh126 treatment extensively decomposed J-aggregates into monomers. Such effect could be greatly reversed by the knocking out of Nur77 (Figure 4D,E) or by blocking the activity of ASK1, JNK, or p38 (Appendix A). These results suggest that Nur77 induction and AKS1 signaling activation are critical for Bkh126 to stimulate ROS production and induce mitochondrial damage.

### 3.4. Bkh126 Induces ASK1-JNK-Dependent Nur77 Mitochondrial Translocation

To explore how Nur77 mediates the ER stress induced by Bkh126 on ROS production and mitochondrial Δψm loss, we explored whether Nur77 could be exported by Bkh126 and whether it could be translocated to the ER or mitochondria. Our results showed that Bkh126 treatment could dose-dependently promote Nur77 nuclear export (Figure 5A). Such effect could be inhibited by ASK1 (GS-444217) or JNK inhibitor (SP600125) but not p38 inhibitor (SB203580) (Figure 5B), indicating that ASK1-JNK signaling contributes to Nur77 translocation. Cytoplasmic Nur77 was not ER-associated but was co-localized with Mito-tracker (Figure 5C), demonstrating that Nur77 was translocated to the mitochondria. Consistently, Bkh126-induced Nur77 mitochondrial translocation was blocked by JNK inhibitor SP600125 (Figure 5C). Thus, Bkh126 stimulated IRE1α-ASK1-JNK signaling to translocate Nur77 to the mitochondria, which is indicative of a critical process leading to the Bkh126-induced loss of Δψm and the induction of apoptosis.

We then determined whether JNK mediated Nur77 mitochondrial translocation by inducing Nur77 phosphorylation, as Bkh126 consistently induced a larger slow-moving band of Nur77, using gel electrophoresis. Indeed, Bkh126 could strongly phosphorylate Nur77 as detected in Nur77 immunoprecipitates purified from HCC cells that endogenously and ectopically expressed Nur77 using anti-p-Ser/Thr antibody (Figure 5D,E). Nur77 phosphorylation by Bkh126 could be reversed by JNK inhibitor SP600125 (Figure 5F). Since Nur77 S95 was a putative phosphorylation site for JNK [31], we mutated this residue. The mutated Nur77 (S95A) was resistant to the Bkh126 induction of phosphorylation (Figure 5G). Thus, our results suggest that Nur77 mitochondrial translocation is mediated via its S95 phosphorylation by Bkh126.

### 3.5. Bkh126 Induces Bax Activation and Mitochondrial Dysfunction

As mitochondrial Nur77 may initiate apoptosis through switching Bcl-2 from a cell protector to a cell killer [32], we determined whether Bkh126 could promote Nur77 interaction with Bcl-2. Nur77 did not interact with Bcl-2 without stimuli. In contrast, Nur77 was co-localized and complexed with Bcl-2 upon Bkh126 treatment (Figure 6A,B). The co-treatment of JNK inhibitor SP600125 inhibited Bkh126-mediated Nur77 interaction with Bcl-2 (Figure 6C). Although p38 was not essential for Nur77 cytoplasmic translocation, it was required for Bkh126 to induce Nur77 interaction with Bcl-2 (Figure 6D).

It was demonstrated that Nur77 interaction with Bcl-2 results in the exposure of Bcl-2 BH3 death domain and the consequent activation of Bax [32]. We, thus, determined whether Bkh126 could activate Bax. Our results showed that Bkh126 strongly stimulated Bax6A7 antigen exposure, which could be inhibited by ASK1, JNK, and p38 inhibitors (Figure 6E).

The Bax-mediated Δψm changes caused by Bkh126 were further analyzed. JC-1 staining showed that Bkh126-induced Δψm loss was impaired when Bax was knocked out by Crispr/Cas9 (Figure 6F,G), indicating that Bax is critical to mitochondrial dysfunction. Taken together, our findings suggest that Bkh126-stimulated ER stress is sensed by Nur77 to open the mitochondrial permeability transition pore (MPTP) and trigger cell killing.

### 3.6. Anti-HCC Activity of Bkh126 In Vivo

The anti-HCC activity of Bkh126 was finally evaluated in in vivo. The male BALB/c nude mice that harbored subcutaneous HepG2 xenografts were subjected to Bkh126 treatment (20 and 40 mg/kg), injected intraperitoneally once daily for 3 weeks. As shown, the tumor growth was dose-dependently inhibited by Bkh126. The treatment of 20 mg/kg Bkh126 reduced tumor volume by ~51% (Figure 7B), while 40 mg/kg shrank the tumor by ~82% (Figure 7A). The anti-HCC activity of Bkh126 mediated by Nur77 was conformed in Nur77/KO xenografts and compared to that of sorafenib, which is currently used in HCC. The tumor shrinkage effect of Bkh126 (−69.6%) was revealed to be superior to that of sorafenib (−48.8%) at the same dosage of 30 mg/kg once daily for 3 weeks (Figure 7B). The knocking out of Nur77 almost completely inactivated the action of Bkh126 but did not significantly impact on the efficacy of sorafenib (Figure 7B). Mechanistically, Bkh126 dose-dependently induced Nur77 expression, which was accompanied with increased phosphorylation levels of IRE1α, ASK1, JNK, and p38 (Figure 7C). Taken together, the underlying mechanism for the anti-HCC activity of Bkh126 was proposed (Figure 7D).

Bkh126 did not affect mouse behaviors and weight gain (Appendix A). HE staining showed that Bkh126 did not significantly induce pathological changes in the liver, lung, heart, kidney, or spleen (Appendix A). Consistently, the in vitro analyses showed that Bkh126 did not interfere with the viability of normal LO2 cells (IC50 > 50 μM), whereas it could dose-dependently suppress the viability of various HCC cells, including HepG2, Huh-7, and Hep3B (Appendix A).

## 4. Discussion

HCC ranks at the top of the causes of cancer-associated death [33]. Although there are various tyrosine kinase inhibitors (TKIs; e.g., sorafenib, lenvatinib, regorafenib, and cabozantinib) and monoclonal antibodies against PD-L1 (atezolizumab plus bevacizumab) currently available for the treatment of advanced HCC, the response rate is low, and the acquired resistance has become an increasing problem [34]. It remains urgent to develop new strategies (novel therapeutic targets and new drugs) to meet the considerable unmet clinical needs. Natural products are important sources in drug discoveries and developments. The naturally occurring sesquiterpenoids are critical therapeutic leads [35]. In this study, we identified cryptomeridiol (Bkh126), isolated from traditional Chinese medicine *Magnolia officinalis* [21,22], as a potent cytotoxic sesquiterpenoid against HCC cells. The anti-HCC activity of Bkh126 is initiated by the aggravating of ER stress and is dependent on Nur77 induction, as the suppression of ER stress and the knockout of Nur77 strikingly reversed the apoptotic and tumor-suppressive effects of Bkh126. The in vivo experiments demonstrated that Bkh126 is superior to sorafenib, which is currently used in advanced HCC treatment.

HCC commonly harbors dysregulated IRE1α signaling [26,27]. IRE1α is increased in HCC [26,27]. ASK1, the primary downstream effector of IRE1α, is implicated in chronic liver diseases, pre-cancerous lesions, and HCC [20,36,37]. XBP1, another downstream of IRE1α is a specific activator of α-fetoprotein (AFP) [20]. It also acts to promote IL-6 release, trigger epithelial–mesenchymal transition (EMT), and promote HCC metastasis [20,27]. Thus, overactive IRE1α signaling is suggested to be implicated in hepatocarcinogenesis. Our results suggest that Bkh126 exploits pre-elevated IRE1α and ASK1 signaling to combat HCC.

It has been well-established that increased ER stress and the chronic activation of the UPR are implicated in tumorigenesis in a wide range of malignancies [16]. However, it appears unsatisfactory to develop UPR inhibitors for therapeutics. On the other hand, the aggravation of the pre-activated UPR has been suggested to be a promising strategy [38,39,40]. Our study described that naturally occurring sesquiterpenoid Bkh126 acts to provoke the pre-activated UPR. Sesquiterpenoid thapsigargin (from *T. garganica*) is another ER aggravating agent recently developed as a prostate-specific antigen-activable prodrug to treat prostate cancer [38]. Tunicamycin has been shown to specifically exacerbate ER stress to overcome chemoresistance in multidrug-resistant gastric cancer cells [39]. A recently identified compound, ErSO, has been pre-clinically demonstrated to completely remove the tumor burden of therapy-resistant breast carcinoma through reinforcing the estrogen receptor (Y537S/D538G)-mediated pre-activated UPR [40]. Thus, the aggravation of the pre-activated UPR has emerged as a promising strategy for cancer treatment, and understanding the underlying mechanisms is critical for the development of new ER-disturbing drugs.

We demonstrated that Nur77, which is frequently downregulated in HCC [41], is rapidly induced by Bkh126 to sense IRE1α-ASK1 signaling. Nur77 is stress inducible and is suggested to be involved in ER stress regulation [42]. However, how Nur77 mediates ER stress-associated cell death remains poorly understood. Nur77 has been described to translocate to ER and exacerbate ER stress for triggering cell death [3,4]. In our study, we did not observe that Nur77 moves to the ER. Instead, the activation of IRE1α-ASK1-JNK signaling by Bkh126 translocates Nur77 to the mitochondria, where it reduces mitochondrial membrane potential (Δψm), causing a deadly effect. It is common for the cross-talk between the ER and the mitochondria [43]. Our study demonstrated that Nur77 mediates Bkh126-induced ER stress deleterious to mitochondrial dysfunction, converting a cytoprotective UPR into a pro-apoptotic UPR.

*Magnolia officinalis* is listed in Chinese pharmacopoeias and contains pharmacologically active components proven to be beneficial for combating different human diseases [44,45]. In our study, Bkh126 revealed to have no significant cytotoxicity against normal cells. It did not significantly affect the mouse weight gain and caused no obvious damage to multiple organs during Bkh126 treatment. Looking into therapeutic leads with potential targets is critical for formulating a modern drug. The target identification for Bkh126 in this study provides a basis for improving the leads for the development of anti-HCC drugs.

## Figures and Tables

**Figure 1 cells-11-03870-f001:**
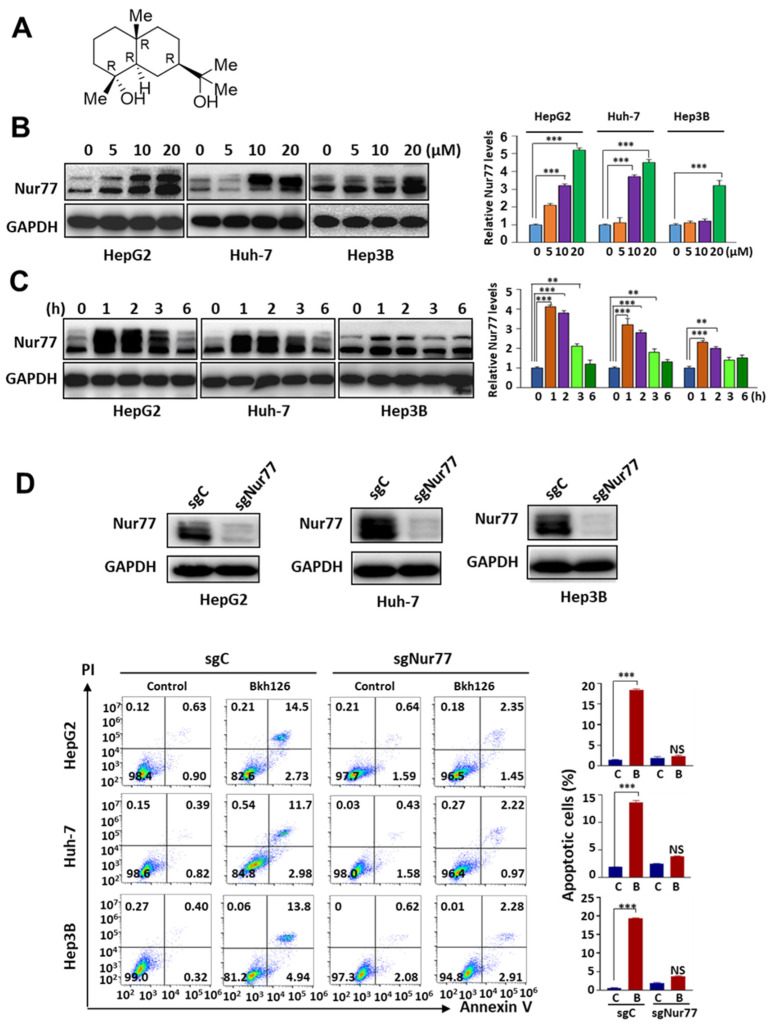
Bkh126 induces Nur77 expression and HCC cell apoptosis: (**A**) Chemical structure of Bkh126. (**B**,**C**) HCC cells were treated with different concentrations of Bkh126 for 2 h or with 20 μM Bkh126 for different time intervals. Vehicle treatments served as controls. Cells were lysed and blotted with anti-Nur77 and GAPDH. (**D**) HCC cells and their Nur77 KO cells were treated with vehicle or 20 μM Bkh126 for 24 h. The apoptotic rates were calculated based on AnnexinV/PI staining. ** *p* < 0.01 and *** *p* < 0.001. C, control; B, Bkh126.

**Figure 2 cells-11-03870-f002:**
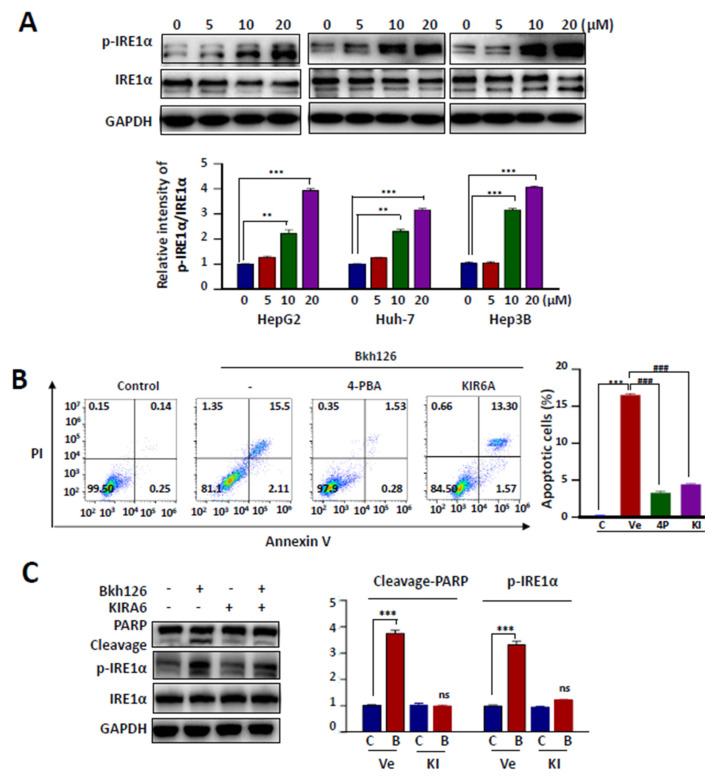
Bkh126 induces ER stress and UPR activation: (**A**) HepG2, Huh-7, and Hep3B were treated with vehicle or Bkh126 (0, 5, 10, and 20 μM) for 24 h. IRE1α and p-IRE1α were determined using Western blotting. The blot intensities were quantified. (**B**) HepG2 cells were treated with 20 μM Bkh126 alone or in combination with 500 μM 4-PBA (ER stress inhibitor) or 5 μM KRIA6 (IRE-1αinhibitor) for 24 h. Flow cytometry was used to assay the apoptotic rates through Annexin V/PI staining. (**C**) HepG2 cells were pre-treated with 5 μM KIRA6 (IRE1α inhibitor) and then 20 μM Bkh126 for 24 h. The protein expression and phosphorylation of IRE1α were blotted. The cleavages of PARP were compared. ** *p* < 0.01, *** *p* < 0.001, and ^###^
*p* < 0.001. ns, not significant; C, control; B, Bkh126.

**Figure 3 cells-11-03870-f003:**
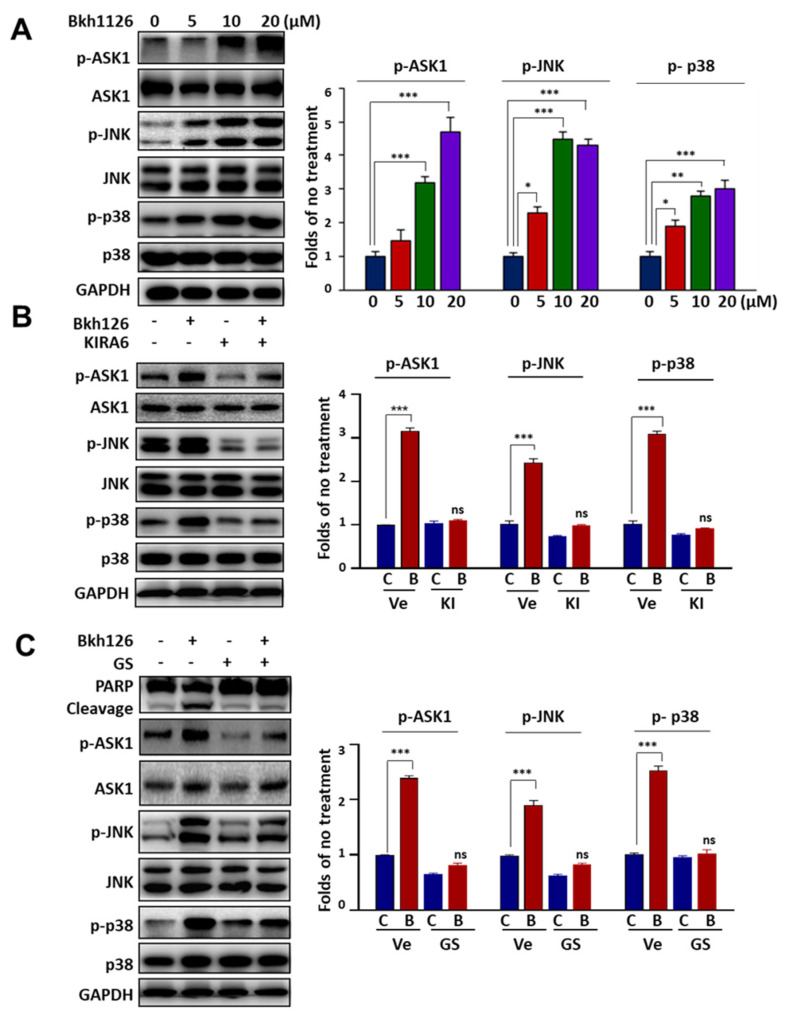
Bkh126 induces IRE1α-ASK1 activation and apoptosis: (**A**) HepG2 cells were treated with increasing concentrations of Bkh126 for 24 h. The cells were lysed and blotted with antibodies against total and phosphorylated proteins of ASK1, JNK, and p38. The protein levels were quantified based on the blot gray intensities. (**B**,**C**) HepG2 cells were pre-treated with 5 μM KIRA6 (IRE1α inhibitor) (**B**) or 1 μM GSK-444217 (ASK1 inhibitor) (**C**) and then 20 μM Bkh126 for 24 h. The protein expression and phosphorylation of ASK1, JNK, and p38 were blotted. The cleavages of PARP were compared among groups. * *p* < 0.05, ** *p* < 0.01, and *** *p* < 0.001. C, control; B, Bkh126; Ve, vehicle; KI, KIR6; GS, GS-444217.

**Figure 4 cells-11-03870-f004:**
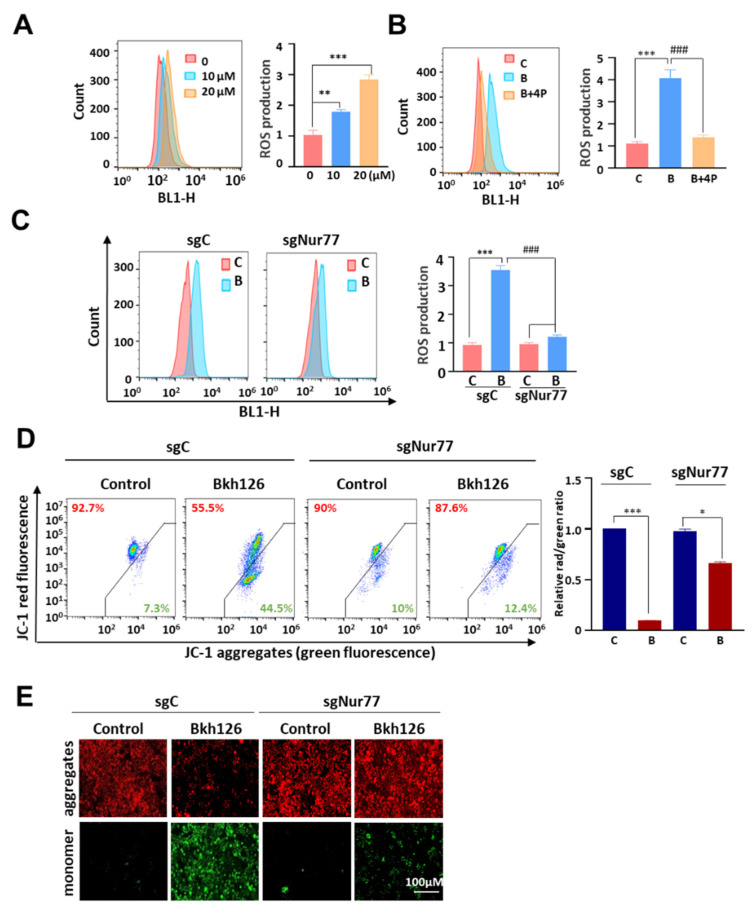
Bkh126 induces mitochondrial membrane potential dissipation: (**A**,**B**) ROS assays. HepG2 cells were treated with increasing concentrations of Bkh126 (**A**) or with 20 μM Bkh126 combined with 4-PBA (ER stress inhibitor) (**B**) for 24 h. The cells were lysed and subjected to ROS analyses. (**C**) HepG2/sgC and HepG2/sgNur77 were treated with 20 μM Bkh126 for 24 h. The cells were lysed and subjected to ROS analyses. (**D**,**E**) HepG2/sgC and HepG2/sgNur77 cells were treated with 20 μM Bkh126 for 24 h. The cells were imaged with a fluorescent microscope (**C**) and analyzed with flow cytometry (**D**). * *p* < 0.05, ** *p* < 0.01, *** *p* < 0.001, and ^###^
*p* < 0.001. C, control; B, Bkh126.

**Figure 5 cells-11-03870-f005:**
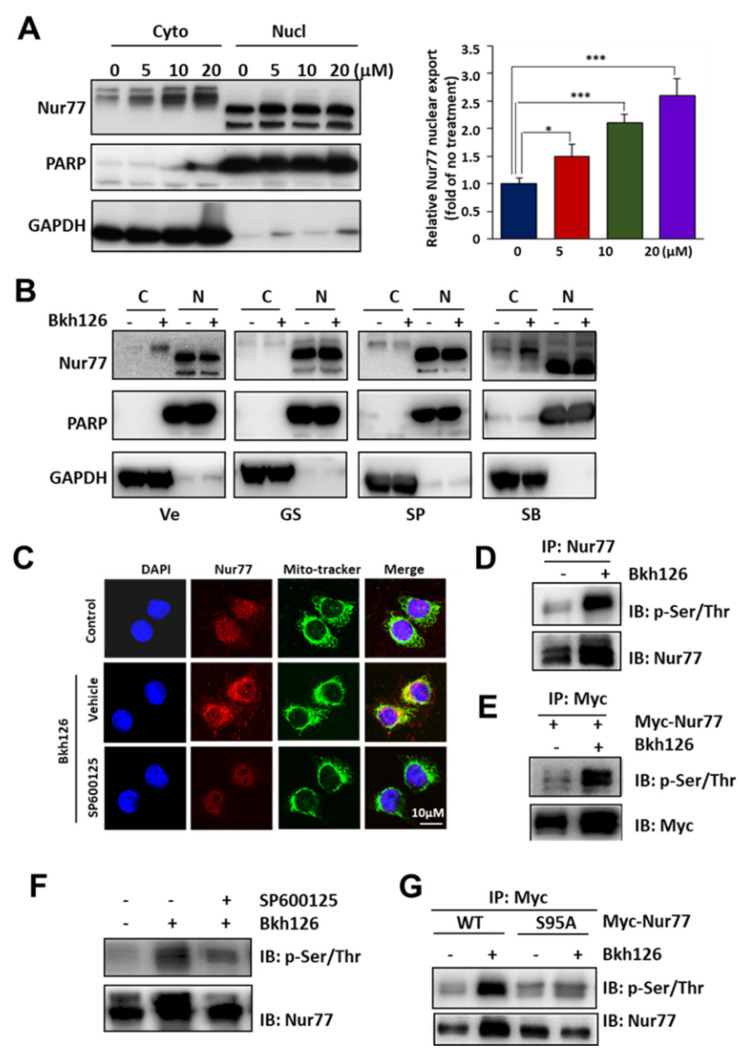
Bkh126 induces Nur77 translocation to mitochondria: (**A**,**B**) HepG2 cells were treated with Bkh126 (0, 5, 10, and 20 μM) for 24 h or Bkh126 (20 μM) in combination with GS-444217 (ASK1 inhibitor), SP600125 (JNK inhibitor), or SB203580 (p38 inhibitor) for 2 h. The lysates were fractionated in cytosol (C) and nuclei (N). The translocation of Nur77 was analyzed. (**C**) HepG2 cells harboring the green fluorescence of Mito-tracker were treated with 20 μM Bkh126 for 2 h and then stained with anti-Nur77 and cy3. Fluorescent images were captured using confocal microscopy. (**D**,**E**) HepG2 cells and HepG2 Nur77 OE stable cells were treated with 20 μM Bkh126 for 2 h. The cells were lysed and immunoprecipitated with antibody against Nur77 (endogenous) (**D**) or Myc epitope (ectopical) (**E**) and blotted with anti-p-Ser/Thr. (**F**) HepG2 cells were pre-treated with SP600125 (JNK inhibitor) for 1 h and then 20 μM Bkh126 for 2 h. The phosphorylation of Nur77 was determined with anti-p-Ser/Thr in Nur77-immunoprecipitated complex. (**G**) HepG2 cells were transfected with wildtype Nur77 or Nur77 S95A and treated with 20 μM Bkh126 for 2 h. The lysates were immunoprecipitated with anti-Nur77 or anti-Myc and subjected to anti-p-Ser/Thr staining. * *p* < 0.05, and *** *p* < 0.001.

**Figure 6 cells-11-03870-f006:**
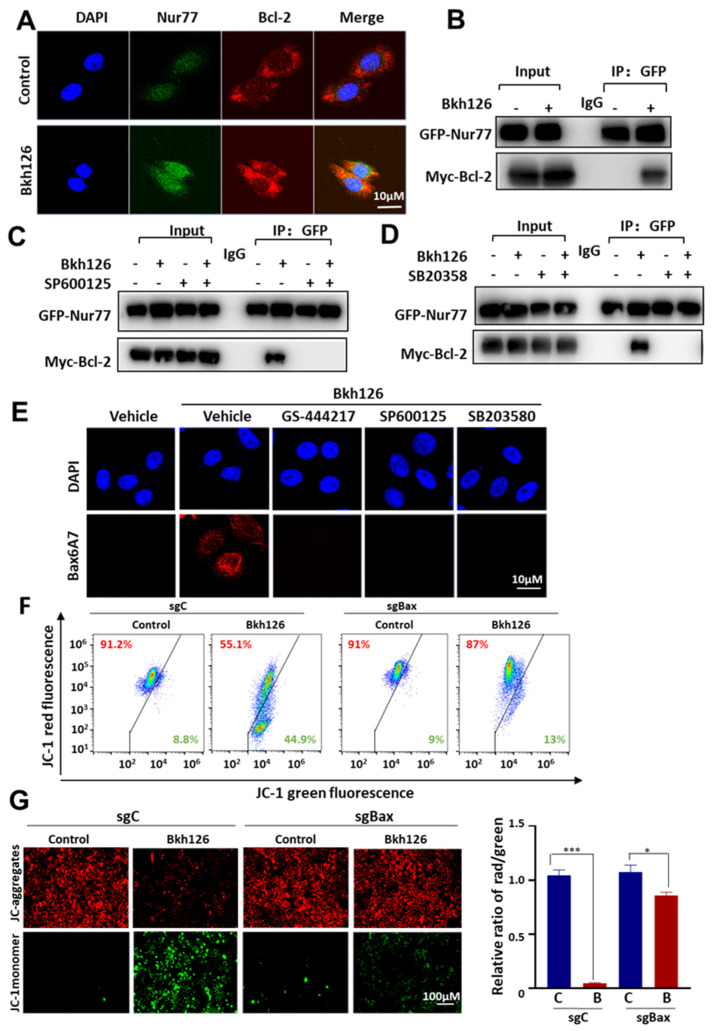
Bkh126 induces Nur77 interaction with Bcl-2 and Bax activation: (**A**) HepG2 cells were treated with 20 μM Bkh126 for 2 h and then stained with anti-Nur77 and anti-Bcl-2. The cells were co-stained with DAPI. (**B**–**D**) HepG2 cells were co-transfected with GFP-Nur77 and Myc-Bcl-2. The cells were treated with 20 μM Bkh126 in the presence or absence of SP600125 (JNK inhibitor) or SB203580 (p38 inhibitor) for 2 h. The lysates were co-immunoprecipitated with anti-GFP and blotted with anti-GFP and anti-Myc. (**E**) HepG2 cells were pre-treated with GS-444217 (ASK1 inhibitor), SP600125 (JNK inhibitor), or SB203580 (p38 inhibitor) and then 20 μM Bkh126 for 2 h. The cells were immunostained with anti-Bax6A7 and co-stained with DAPI. Fluorescent images were captured using confocal microscopy. (**F**,**G**) HepG2/sgC and HepG2/sgBax cells were treated with 20 μM Bkh126 for 24 h and analyzed with JC-1 staining. The cells were examined with a fluorescent microscope and flow cytometry. * *p* < 0.05 and *** *p* < 0.001. C, control; B, Bkh126.

**Figure 7 cells-11-03870-f007:**
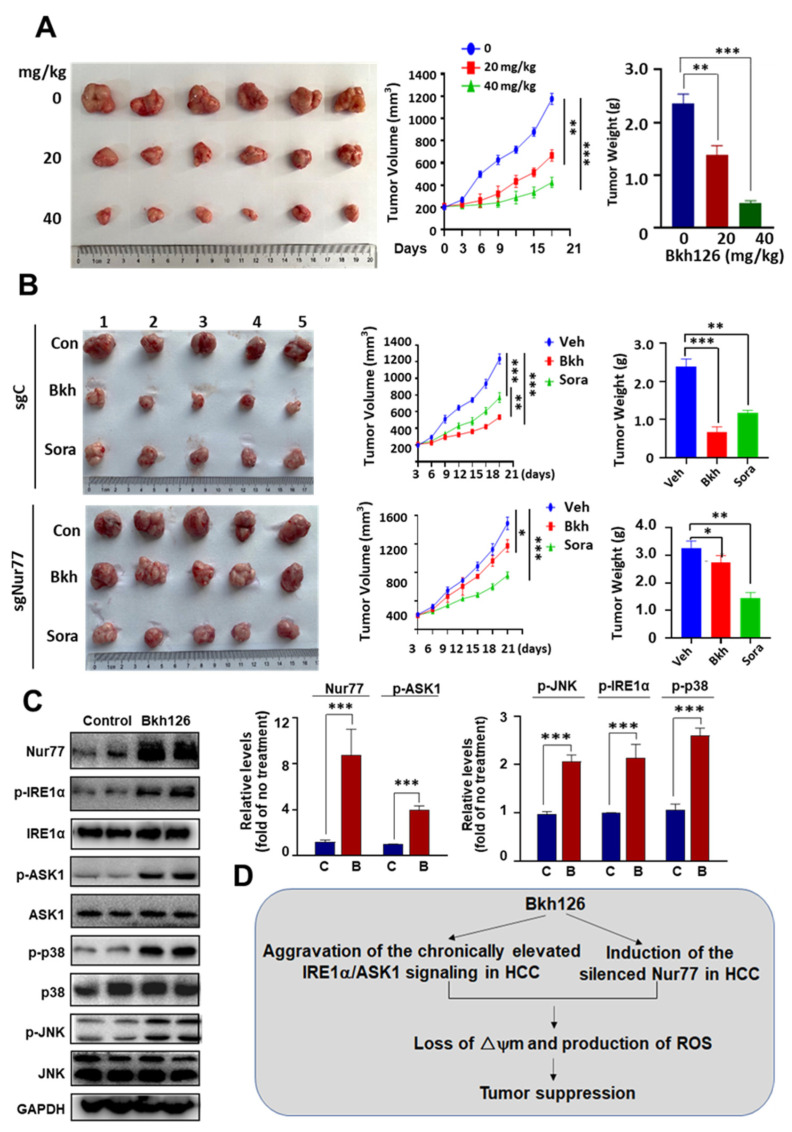
Bkh126 suppresses HCC xenograft tumor growth: (**A**) Male BALB/c nude mice harboring HepG2 xenograft tumors were treated with vehicle or Bkh126 (20 or 40 mg/kg) intraperitoneally once daily for 3 weeks (n = 6 per group). (**B**) Male BALB/c nude mice harboring HepG2 or HepG2(sgNur77) xenografts were intraperitoneally treated with vehicle, Bkh126 (30 mg/kg), or sorafenib (30 mg/kg) intraperitoneally once daily for 3 weeks (n = 5 per group). In (**A**,**B**), tumor volume was measured with a caliper every three days. The tumors were photographed and weighted. (**C**) Nur77 induction by Bkh126 was determined. The expression and phosphorylation of ASK1, JNK, and p38 were compared between vehicle and Bkh126 treatment (20 mg/kg). (**D**) Mechanistic scheme. * *p* < 0.05, ** *p* < 0.01, and *** *p* < 0.001. C, control; B, Bkh126.

## Data Availability

Not applicable.

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
