# Peer review of "Orphan Nuclear Receptor Nur77 Mediates the Lethal Endoplasmic Reticulum Stress and Therapeutic Efficacy of Cryptomeridiol in Hepatocellular Carcinoma"

_cells, 2022, doi:10.3390/cells11233870_

Round 1

Reviewer 1 Report

The authors evaluated the effects of Bkh126 on HCC via Nur77 and mediating IRE1α-ASK1 16 signaling. The study is interesting and can add to the field knowledge. The authors would like to see the following comments addressed first:

1) rescue experiments would be needed to establish the Nur77 expression induced by Bkh126 after the knockout;

2) there are general overclaims about the safety data of Bkh126. For instance, in Line 363 - 364, the authors wrote 'the safety of Bkh126 was also evidenced that it did not exert significant toxicity to liver, lung, heart, kidney and spleen. But in Supplementary Fig. S3B, only HE staining was done. More extensive animal tox studies will be needed for such claims. Otherwise, the authors are advised to revise/tone down the relevant statements;

3) sorafenib was referred to as the first-line of HCC, when new treatment has emerged (atezolizumab + bevacizumab);

4) there are grammatical and spelling errors. Please perform thorough proofreading.

Reviewer 2 Report

This investigation was well-executed and show novel evidence on the effect of a natural extract, namely cryptomeridiol, on liver cancer both in vitro and in vivo. Some minor issues still need to be addressed.

Authors should explain/discuss about the doses/vias that people use when consume M. officinalis extract to give a full picture of cryptomeridiol to the readers.

For an easier reading of the figures, define in the legend of figures 1, 2, 3, 4, 6 and 7 the meaning of “C” and “B” shown in x axis of some plots.

The manuscript contains some typographical errors that need to be corrected; for example, in discussion section: Natural “pro00ducts” are important sources…

It is not necessary to include “*p<0.05” in the legend of figure 1 since it was not indicated in the figures; the only indicated p-value were “**p<0.01 and ***p<0.001”. In figure legend of figure 4 it is also unnecessary to include “**p<0.01” since was not used, but “*p<0.05” should be included in the legend of figure 6 since it is shown in the figure.
